# A 3D-Printed Dummy for Training Distal Phalanx Amputation in Mice

**DOI:** 10.3390/ani14081253

**Published:** 2024-04-22

**Authors:** Miriam Heuser, Fernando Gonzalez-Uarquin, Maximilian Nuber, Marc A. Brockmann, Jan Baumgart, Nadine Baumgart

**Affiliations:** 1Translational Animal Research Center, University Medical Centre, Johannes Gutenberg-Universität Mainz, 55122 Mainz, Germany; fernando.gonzalez@uni-mainz.de (F.G.-U.); maximilian.nuber@unimedizin-mainz.de (M.N.); jan.baumgart@uni-mainz.de (J.B.); nadine.baumgart@uni-mainz.de (N.B.); 2Clinic and Polyclinic for Neuroradiology, University Medical Centre, Johannes Gutenberg-Universität Mainz, 55131 Mainz, Germany; mbrockma@uni-mainz.de

**Keywords:** alternative methods, distal phalanx amputation, education, mouse marking, reduction

## Abstract

**Simple Summary:**

In scientific research, there are various fields that require the simultaneous sampling of tissue and marking of mice shortly after birth. One approach is the amputation of the distal phalanx, a method that not only provides a lasting mark but also allows for the use of the removed tissue for early genotyping. While the use of this method is a topic of ongoing debate, it remains one of the safest and most widely employed techniques in current practice. The key lies in practical experience, ensuring the correct application of the method. To enhance and optimize this process, we have developed a 3D dummy specifically designed for training individuals in the precise amputation of the distal phalanx in 5-day-old mouse pups. In a recent seminar, we sought to evaluate the effectiveness of training on this innovative model and assess its appropriateness for the task at hand.

**Abstract:**

The development of realistic dummies for training the distal phalanx amputation (DPA) technique in mouse pups is a promising alternative to reduce and replace animals in training for research and teaching. To test this, we obtained micro-CT data from postnatal day-five mouse pups, meticulously segmented them, and converted them into a 3D mesh format suitable for 3D printing. Once the dummy was printed, it was evaluated during actual training courses in two different groups: in the first group, users received no dummies to train the DPA, and in the second group, users were trained with three dummies. To assess the effectiveness of the dummy, we conducted a survey followed by an expert veterinarian evaluation. Our results showed that DPA is a complex procedure, and it is commonly poorly performed. When implementing the dummies, users who were not provided with dummies to practice only had an 8.3% success rate in DPA, while users provided with three dummies had a 45.5% success rate, respectively. Despite additional research being needed, our dummy offered improved practical training by providing a safe and effective alternative in line with ethical considerations while demonstrating the feasibility of using 3D printing technology to promote the 3Rs in experimental research.

## 1. Introduction

In biomedical experiments, it is common to house mice in groups for practical and welfare reasons [1,2,3]. Experimental designs often require a minimum number of mice per treatment, needing multiple groups to evaluate various treatments to ensure replicability. Housing several mice per cage requires selecting an appropriate method for individual mouse marking. An optimal marking method should provide an individual with permanent identification while minimizing distress and the occurrence of diseases [4].

For certain research projects, especially with genetically altered animals, animal caretakers and scientists need to mark mice a few days after birth, with simultaneous collection of tissue for genotyping. Distal phalangeal amputation (DPA) is a widely used protocol for marking mouse pups shortly after birth, which also allows for the collection of tissue. In this procedure, the distal phalanx (2 to 3 mm) of a toe is removed from the mouse pups [5]. The timing of the procedure is critical, as performing it before postnatal day 5 (P5) leads to additional mistakes in the DPA (the toes are not fully separated), while performing it after P9 increases the risk of pain during and after the procedure [6,7]. Studies reported that optimal results are achieved when the distal phalanx removal is performed between P5 and P7, as it does not have adverse physiological or behavioral effects on the mice [6,8]. The advantages of this method are its permanence over time and the fact that the removed tissue can be used for genotyping [9]. Adult mice can be identified by the missing tips on their paws [5].

Precise amputation of the distal phalanx is a mandatory prerequisite. This requires a high level of skill on the part of the staff. Otherwise, the holding power of the animal can be impaired if the amputation is too extensive. If, on the other hand, too little is cut off, the claw can grow back and the marking can no longer be recognized as the animal matures. Furthermore, inadequate DPA training causes distress and anxiety in caretakers, researchers, and animals [10,11]. For example, the fact of dealing with young animals that are still naked and blind at this time, but are quite capable of making sounds, can be stressful and emotionally distressing for the user (word-of-mouth communication). Moreover, removing young animals from the nest causes stress for the mother and fear of death for the mouse pups. On a larger scale, education and training added to research, testing, routine production involved approximately 3,879,600 mice in the European Union in 2020 [12]. Given that, the following question arises: How can users be effectively trained in DPA while adhering to the principles of the 3Rs (replace, reduce, and refine)? [13].

It is a matter of fact that DPA is a topic of ongoing debate. Nonetheless, if it is performed well by trained personnel, it may be one of the safest and most useful techniques in current practice. The European Directive 2010/63/EU urges using alternatives to animal experiments whenever possible [14]. Developing and implementing animal-based dummies can create a safe environment for training, reducing unnecessary harm to animals and minimizing the number of animals used in training [15,16]. The validation of animal-based dummies in veterinary education and surgical protocols has demonstrated benefits in promoting the 3Rs [17,18,19]. The use of dummies has become an established part of modern teaching in veterinary training (e.g., for learning endodontics in dogs [20]) and human medicine (e.g., for learning anatomy) [21]. However, animal-based dummies have limitations, such as a lack of biological diversity and the tendency to represent real organisms inaccurately [22,23,24]. Developing realistic animal-based dummies is crucial for achieving genuine reduction and replacement.

Three-dimensional printing (3D printing) has presented significant advancements in recent decades, allowing for rapid prototyping of highly complex organisms [25]. This technology enables advanced preoperative planning and haptic simulation-based alternatives for surgical planning and education, including applications in medicine [26,27,28] and also in veterinary medicine [29,30].

This study aimed to create a 3D-based postnatal day-5 mouse dummy for DPA training in order improve amputation skills without causing suffering to animals. To achieve this, we obtained and segmented micro-CT data and converted them into a 3D mesh format for printing. Once printed, we evaluated the dummy based on user perceptions and expert veterinarian evaluation. We hypothesize that our dummy would allow the acquisition of procedural skills complying with optimal training skills and the 3R principles.

## 2. Materials and Methods

### 2.1. Mice

This study was carried out according to the German Animal Welfare Act guidelines and the European Directive 2010/63/EU for protecting animals used for scientific purposes. Reporting was carried out following the ARRIVE guidelines [31]. The offspring (*postnatal* 5 days—P5) were sacrificed by expert personnel for scientific use according to §4 of the Animal Welfare Act, legalized and controlled by the Animal Welfare Authority of the Medical University.

Term-pregnant C57BL/6JRj mice were purchased (Janvier Labs, Le Genest-Saint-Isle, France). After arrival, the mice were housed in type II long filter-top cages (SealSafe Plus, polyphenyl sulfone, 365 mm L × 207 mm W × 140 mm H, Greenline; Tecniplast, Buguggiate, Italy). The mice were kept in a 12:12 h light/dark cycle (lights 06:00–18:00) in a temperature- and humidity-controlled animal room (22 ± 2 °C, 55 ± 5%). Food (ssniff M-Z Extrudat, ssniff, Soest, Germany) and water were supplied ad libitum.

The pup was sacrificed with sodium pentobarbital (i.p. 0.1 mL/10 g BW, Narcoren^®^ (16 g pentobarbital sodium 16.0 g, benzyl alcohol 3.0 g; Boehringer Ingelheim, Ingelheim, Germany)) at P5 for micro-computed tomography.

### 2.2. Micro-Computed Tomography

After euthanasia, the pups were kept in a fixative solution (Roti Histofix^®^, Carl Roth GmbH + Co. KG, Karlsruhe, Germany) for 48 h; the Histofix^®^ was renewed after 24 h. The following day, the embryos were relocated into phosphate-buffered saline (PBS tablets, 1x concentration, pH = 7.45 Gibco Thermo Fisher Scientific, Waltham, MA, USA). Subsequently, the pups were dyed (15% ‘Lugol’s Solution) overnight. CT analysis was performed after incubating in 1x PBS overnight again.

CT was performed using a microfocus X-ray system (Cheetah EVO YXLON GmbH, Hamburg, Germany). The voltage and current parameters were individually set per sample to achieve an optimal contrast (number of projections: 710; scan time: 318 s; voltage: 105 kV; current: 64 µA). Raw data were processed with VG Studio Max (VG Studio Max 3.2, Volume Graphics, Heidelberg, Germany) software and rendered as an STL file.

### 2.3. Dummy Creation

The final 3D image of a P5 pup obtained from the micro-CT “FF35CT” (Yxlon international GmbH, Am Walzwerk 41, 45527 Hattingen, Germany) was printed by Alphacam GmbH (Erlenwiesen 16, 73614 Schorndorf, Germany) using the Stratasys J850 pro 3D printer according to their internal procedures. For the 3D dummy set-up, the printing utilized poly jet technology with a layer thickness of HM-HS and a resolution of 600 × 600 × 1800 dpi. The rubbery material used for printing was Agilus 30 white, and the print mode was set to matte. The support material was subsequently removed, resulting in the final 3D dummy of the P5 pup (Figure 1). The accuracy of the Stratasys J850 Pro 3D printer has a typical deviation from STL dimensions for models printed with rigid materials, based on size: under 100 mm ± 100μ; above 100 mm ± 200μ; or ±0.06% of part length, according to the manufacturer (Stratasys, Ltd., Edina, MN, USA).

### 2.4. Course and Survey

The training course was conducted to assess the practicality of learning the distal phalanx amputation. Due to the availability of the participants, we could not base our results on the size of the sample, but on the largest number of people willing to participate in the course. The participants were randomly assigned to the control or experimental group based on the chronological order in which they signed up to attend our course. The participants included certified and trainee animal caretakers, medical technical assistants, postgraduate students, and other scientific assistants. The course consisted of a theoretical component, where the participants were presented with the legislative background and the amputation technique, including potential sources of mistakes. To avoid confounding factors, all participants were given the same dummy model, the same materials, and equal time to perform the amputation.

Following the theoretical part, a practical exercise was conducted. The participants were randomly divided into two groups. Group one received no dummies for practicing the DPA (representing the current situation where animal caretakers have to practice amputation on living animals directly). Group two received three dummies (54 toes to practice). Participants from group two were instructed to practice the DPA on all four paws of the dummy, targeting all toe-end phalanges using fine scissors, as explained in the presentation before. Each participant was given sufficient time for the exercise. Subsequently, all participants were provided with a decapitated 5-day-old pup. The P5 juveniles (different genetic strains without burden phenotype) for the control were killed for the training courses by decapitation by trained personnel. The animals were frozen until used for course evaluation. The mother mice were reused for further research projects. The task was for each participant to mark a specific number, following a predetermined numbering scheme (Figure 2), on the dead pup by performing the distal phalanx amputation. The revised scheme now focuses on amputating a single toe-end glide per paw, with smaller numbers primarily allocated to the rear paws. As the thumb toe of the hind paw is extremely small, adequate amputation of the distal phalanx on this toe cannot be ensured. Therefore, in numbers 0 and 10, unfortunately, the amputation of two toe-end phalanges had to be resorted to. This allocation ensures that, in the optimal scenario, the marking is predominantly performed on the rear paws, considering the importance of correct grasping for the mice. To ensure fairness and enable the best possible comparison, we decided that all participants should use the same numbering scheme for marking. We assigned numbers for each paw: front left: 40; front right: 70; back left: 4; back right: 8 (the number under which the animal would be found in the system is the sum of the four numbers: 122).

### 2.5. Survey and General Content of the Experiment

We requested all participants to provide feedback by filling out an anonymous evaluation sheet (Table 1). This evaluation aimed to assess the practicality and accuracy of the practice dummy. We evaluated the amputation of the *distal phalanx* based on the correctness of the numbering and incision. The scale ranged from 1 to 3, with the following criteria: 1 = insufficient amputation; 2 = correct amputation; and 3 = excessive amputation. We also evaluated the correctness of the toe selection and identified the correctness of the DPA per animal in each group.

### 2.6. Data Analysis

Data analysis and graph plotting were performed using Prism version 9.5.1 for Windows (La Jolla, CA, USA). We evaluated the relationship between the number of dummies used for training and the success of the DPA through simple logistic regressions. Logistic regression did not require a linear relationship between the dependent and independent variables. Regarding paw preference and performance, we established a threshold where the *p*-value is less than or equal to 0.05, indicating differences between the observed and expected values. For that, we conducted a 2 × 1 arrangement chi-square test. This test was performed using the RStats calculator provided by the State University of Missouri (www.missouristate.edu/rstats; accessed on 18 October 2023). Finally, to compare the user experience in terms of dummy realism, we assigned a quantitative score to user comments given by our professional in animal science. With these data, we conducted a Mann–Whitney U test (a non-parametric test used to compare two groups under the condition of no normality). A total of 24 participants (12 per group) took part in the course. One participant was excluded since they did not correctly respond to the questionnaire. The evaluation of the DPA was carried out on the decapitated pups, so that the authors were not aware of the participants’ group affiliation. However, for data analysis, the authors were aware of the grouping.

## 3. Results

### 3.1. Users’ Perceptions about the Distal Phalangeal Amputation

The overall description of the participants (depicted in Figure 3) revealed that they had experience working with mice (91%). However, this proportion was smaller when referring to experience with <P7 animals (61%) and even smaller when referring to practical experience in the DPA (17%). Surprisingly, the participants revealed a relatively balanced concern about the ethical consideration of the DPA (43%) and a lack of knowledge regarding using the distal phalanx for genotyping (17%). The needs of participants were satisfied by the number of dummies available for practice before conducting the DPA on the dead pups. As mentioned, the participants included certified and trainee animal caretakers, medical technical assistants, postgraduate students, and other scientific assistants; however, the low number of participants did not allow for any categorization.

### 3.2. Users’ Performance on the Distal Phalangeal Amputation

We evaluated the procedure based on the correctness of the amputated distal phalanx. We evaluated 23 amputations performed on the dead mouse pups. Out of the 23 participants whose amputations were assessed, only 9 correctly amputated the distal phalanx on all four paws, receiving a score of “two”. This implies that only 6 out of 23 participants, or approximately 26.1%, achieved the desired outcome of amputating the distal phalanx properly on all paws. In this regard, approximately 73.9% (17/23) of the participants performed errors in amputating one or more distal phalange (Figure 4A).

When analyzing the groups, participants with dummies achieved a success rate of 45.5%. In comparison, those with no dummies had success rates of 8.3% (Figure 5A), which means that for every success in the group with zero dummies, approximately four successes were made in the group with dummies. The comparison using logistic regressions revealed a notable difference between the group with no dummies and the group with three dummies (*p*-value 0.06) (Figure 5B). The lack of statistical significance (if we set a *p*-value < 0.05) may be attributed to the sample size, which can influence regression outputs.

We also checked whether the amputations were performed on the correct toe. The results were similar in both groups, with rates of error of 33.4% and 33.7% for the groups with zero and three dummies, respectively (Figure 5C). After an overall comparison using logistic regressions, we found no relationships between the number of dummies used for practicing and the correctness of the toe selection (Figure 5C,D). Of the total of 23 participants, 3 amputated the distal phalanx on the wrong toe. Five animals even had the wrong number amputated on two paws. Consequently, 8 of the 23 animals received a wrong number for marking. Overall, 34.8% of the animals could not be allocated to the assigned number in retrospect.

The responses to the question regarding which of the paws (front or hint) the participants found the DPA easiest to perform on are displayed in Figure 6A. In total, 55.6% of the participants indicated that they found the DPA easier to perform on the rear paws compared to the front paws, which accounted for 44.4% of the participants. Figure 6B illustrates the distribution of successful DPA results across the front and rear paws, categorized by different dummy groups. The data reveal that participants from the group utilizing three dummies on each of the four paws exhibited the highest success rate. This group accomplished over 60% accuracy in correctly performing DPA on all four paws, and three out of the four paws overcame the threshold established for significance under the 2 × 1 chi-square test. The group without dummies failed to achieve a 50% success rate, specifically on the front left (FL) and rear right (RR) paws.

Participants’ perceptions of the dummy’s realism indicated that the dummy was realistic for most of them (Figure 7). Overall, the users rated the dummy positively concerning its size, weight, and material. Our dummy weighs 4 g and measures 3 cm in length, which is highly comparable to the size and weight of a 5-day-old live mouse. According to the users, another worthy aspect is the advantage of the dummy in improving handling sensitivity while reducing the nervousness and stress of the procedure, particularly for initial training purposes and for beginners engaging in DPA. However, certain aspects, such as the dummy’s flexibility and color, received criticism from the participants.

## 4. Discussion

To the best of our knowledge, our team has pioneered developing and applying a 3D dummy for training distal phalanx amputation, eliminating the necessity of using live or dead mice for procedural training. Our research demonstrates the potential effectiveness of employing dummies as viable alternatives to live animals in educational and research settings. Such findings contribute to advancing strategies aligned with the increasing societal emphasis on animal welfare and the imperative for ethical and humane research practices.

Our dummy helps us to find ways to provide high-quality training in animal-free environments, at least in the first stages of training when animal use is unavoidable (e.g., for marking purposes). Within marking methods, DPA provides a long-life marking system, plus it offers us enough material for genotyping [32]. Nevertheless, as our results showed, the relevance of optimal DPA training lies in the difficulties associated with mastering the procedure. DPA is, indeed, a complex procedure in terms of accuracy. Individuals responsible for marking animals, such as animal keepers or scientists, learn the technique through hands-on experience, which can lead to different mistakes in the procedure (Figure 4B). When DPA is performed wrongly, two main consequences are possible: (1) a risk of insufficient tissue removal during the amputation emerges, resulting in tissue regeneration and compromise of the effectiveness of permanent marking, or (2) a risk of excessive amputation, which can be painful, stressful, and significantly severe, while concomitantly compromising the experimental data. The performance of DPA is approved in that it causes minor stress to the pup. However, if more tissue is removed than prescribed, the minor degree of stress can no longer be assumed.

Aiming to overcome such risks, which might be associated with either a lack of adequate technical training [33] or users’ experience and stress of training directly with mice [34,35,36], we developed and assessed in practice our 3D-dummy to rehearse the DPA.

The results from this study support the idea that dummy-based training might allow researchers to train the DPA in an animal-free way as a strategy for optimizing hands-on skills while complying with the 3Rs. In this regard, the existing literature suggests that participants who had the opportunity to utilize training dummies exhibited enhanced reliability in both animal handling and surgical procedures [17,24,37,38]. Furthermore, the advent of 3D-printing technology enables the creation of dummies with realistic haptic properties, thus facilitating the set-up of realistic procedures. Some examples of the application of 3D printing in rodents are focused on imaging, neurosurgical training [39,40], skeletal and joint defects [41,42], and tissue implantation [43]. Our results emphasize the significance of incorporating dummies to familiarize oneself with the procedures and gain practice during the initial stages of training. These underscore the importance of integrating dummies into training programs for optimal skill development and implementing the 3Rs either by reducing the number of live animals on which the training must be practiced or by reducing the potential stress and pain caused to the animals due to a lack of staff training regarding mistakes in the procedures. The fact that experimenters can refresh and improve their skills without the need for additional animals is a win–win for animal and human welfare.

Three-dimensional models have the potential to become integrated into LAS training programs worldwide, contributing to more ethical and high-quality science in animal research. In the medium- and long-term, cooperation is crucial to avoid models being developed in parallel at different institutions. Thus, creating a central database for sharing image data and 3D print models would be desirable. This would allow easier access for all those who do not have access to infrastructure such as CT or corresponding processing programs. Three-dimensional printing using the latest technology is possible, as in our example, via commercial providers.

While this study has provided valuable insights, we must acknowledge limitations. Firstly, the sample size used in this study may restrict the generalizability and robustness of the findings, particularly in the context of logistic regression tests. Secondly, this study primarily focused on the technical aspects of the amputation procedure and did not directly examine the mechanical properties of the dummy or the potential stress experienced by the animals involved. To address these limitations, future research endeavors should consider larger sample sizes to enhance statistical power and investigate the training program’s impact on technical and mechanical properties while also considering animal welfare. Moreover, developing advanced printing materials and training dummies could significantly improve the realism and effectiveness of the training process.

## 5. Conclusions

Despite debate around DPA’s implementation, it remains a widely effective method for marking mice at an early age, if it is carried out correctly and reliably. This study emphasizes the importance of accurate training for *distal phalanx* amputation in newborn mice to achieve permanent marking. The use of realistic 3D-printed dummies has shown the potential to reduce errors during subsequent amputations on deceased mouse pups. This offers long-term implications for standard training protocols and aligns with ethical considerations in animal research, which means we can reduce the number of animals for training purposes and the user’s stress during the amputation procedure while ensuring precise and effective permanent marking.

## Figures and Tables

**Figure 1 animals-14-01253-f001:**
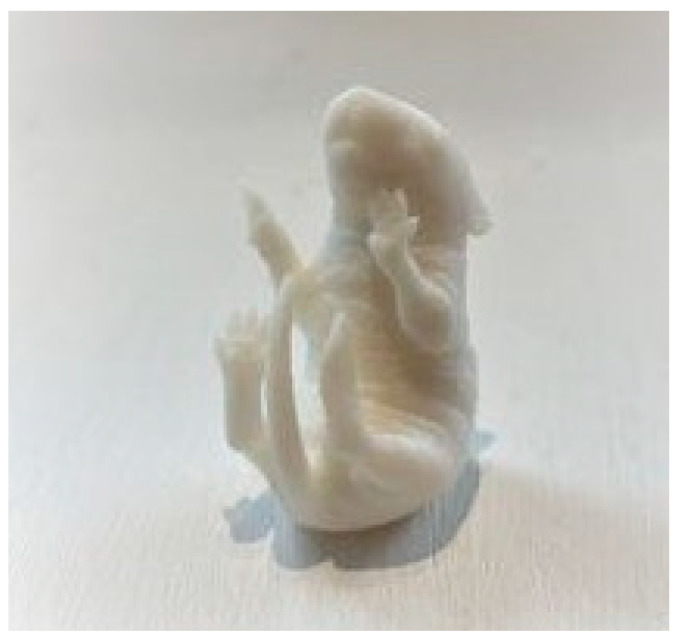
Three-dimensionally printed dummy of the P5 mouse pup. The dummy has a length of 2.7 cm, the widest part between the back and the tail is 1.8 cm, and the narrowest part between the neck and the front paw is 1.2 cm.

**Figure 2 animals-14-01253-f002:**
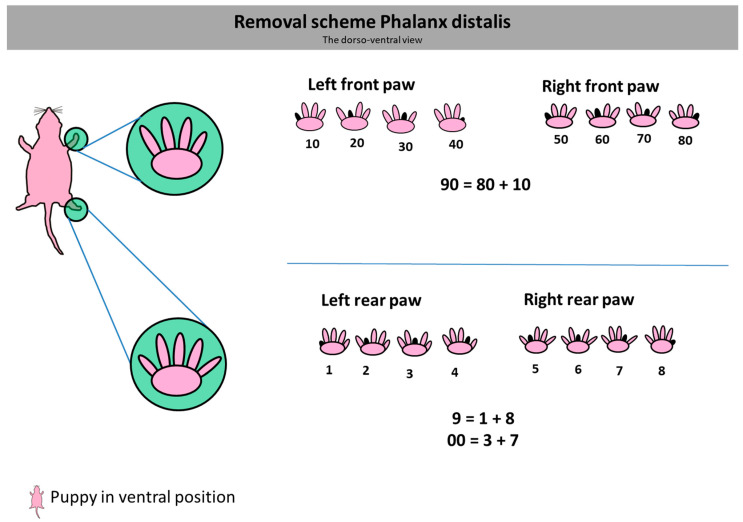
Original schemes developed for the courses. The pictured pup is shown in abdominal position = dorsoventral view. The left front paw represents the numbers from 10 to 40, and the right front paw presents the numbers from 50 to 80. The single-digit numbers are presented on the rear paws (**left** 1–4) and **right** (5–8). As the inner toes on the rear paw are too small to ensure adequate amputation of the front 1/3, these two toes are not included in the scheme. In order to be able to represent the numbers 0 and 9 nevertheless, these were amputated by amputating two toes 9 = 1 + 8 and 0 = 3 + 7. Similarly, the number 90 is obtained by amputating the toes of the numbers 80 + 10.

**Figure 3 animals-14-01253-f003:**
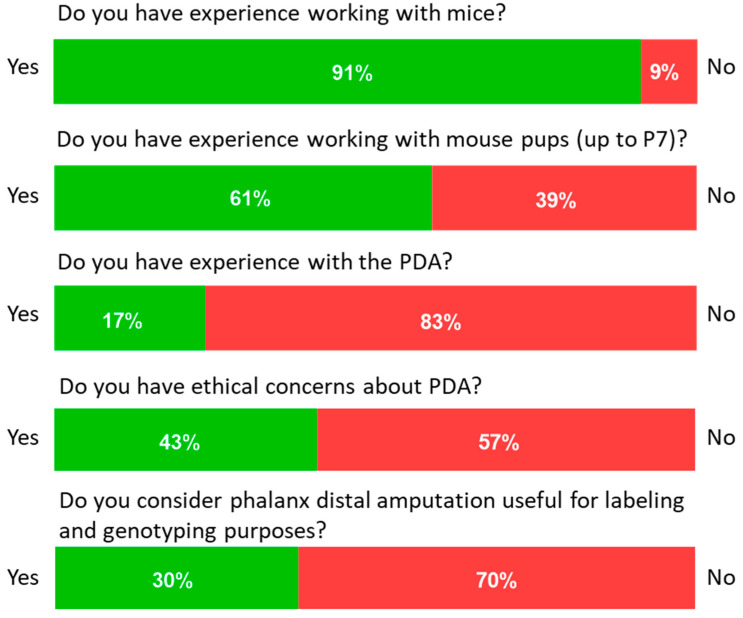
Questionnaire responses on the perceptions toward the DPA.

**Figure 4 animals-14-01253-f004:**
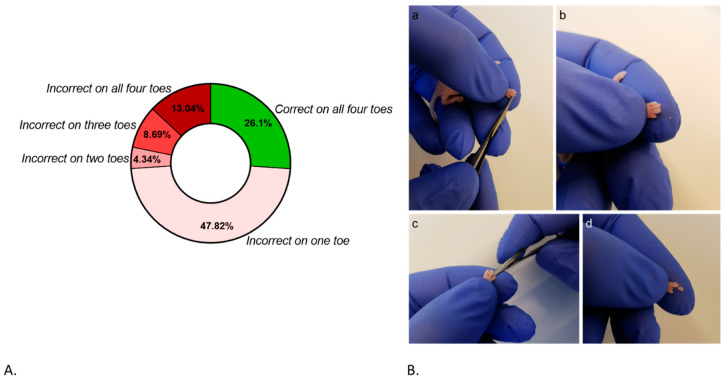
(**A**) Percentage of participants who performed the DPA correctly and incorrectly overall. (**B**) Demonstration of inappropriate amputation of the distal phalanx: (**a**) shows how the scissors are applied to the DPA, and the area to be amputated is not completely covered; (**b**) shows the result from photo a, and the amputated material can be seen on the fingertip of the person performing the procedure; (**c**) shows how the scissors are applied too far back on one toe; (**d**) shows how the tissue removed was the entire toe. This amputation is definitely too far back and critical from an animal welfare point of view.

**Figure 5 animals-14-01253-f005:**
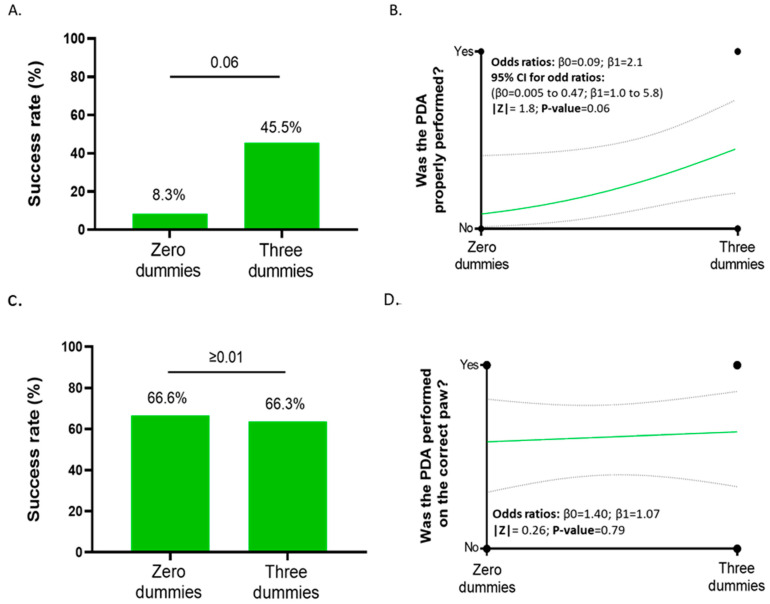
Evaluation of the DPA accuracy on the pup according to the number of dummies used for training: (**A**) The success rate of the DPA from users using zero and three dummies. *p* values represent simple logistic regressions between groups. (**B**) Curve from the simple logistic regression between users training with zero and three dummies. Evaluation of the right toe selection according to the number of dummies used for training. (**C**) The success rate in the paw selection of users using zero and three dummies. *p* values represent simple logistic regressions between groups. (**D**) Curve from the simple logistic regression between users training with zero and three dummies.

**Figure 6 animals-14-01253-f006:**
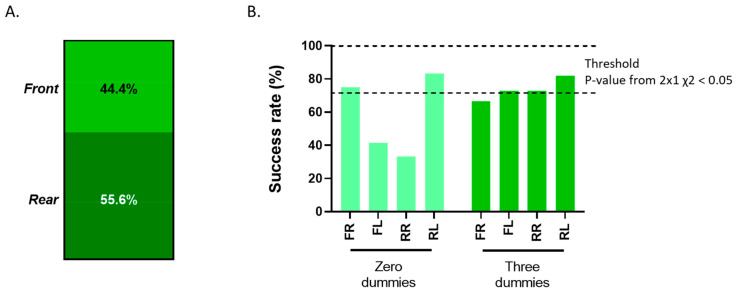
Evaluation of the pup according to each paw: (**A**) Users’ perception about which paw was the easiest to perform the DPA on. (**B**) The success rate of the DPA per paw in each group (zero or three dummies). FR: front right; FL: front left; RR: rear right; RL: rear left. The threshold indicated the range in which the observed significantly differed from the expected based on the chi-square test.

**Figure 7 animals-14-01253-f007:**
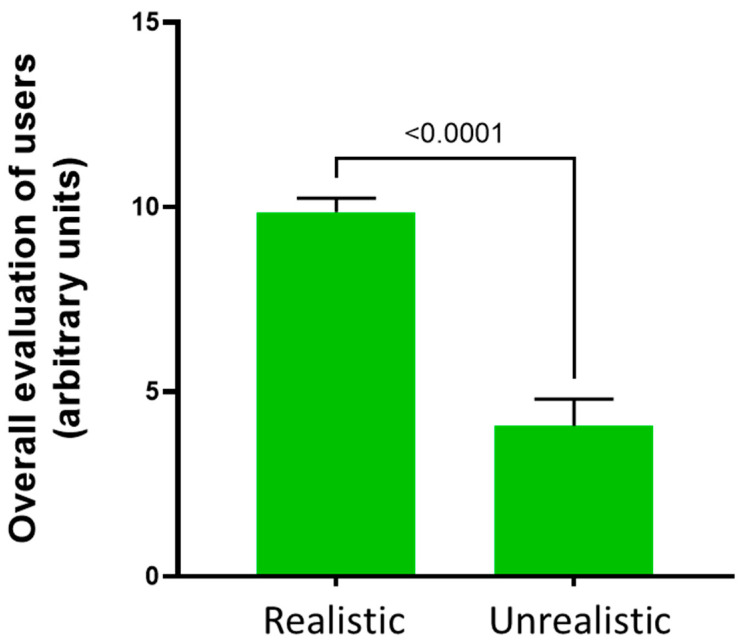
Evaluation of the dummy’s realism according to the users’ perceptions. Statistical comparisons were performed using the Mann–Whitney U test. Bars indicate standard deviation (SD).

**Table 1 animals-14-01253-t001:** Questionnaire to the users about their experience with DPA.

Question	Answer	
Have you ever worked with mice?	Yes (1)	No (0)
Have you ever worked with young animals (Up to p7 = 7 days post-partum)?	Yes (1)	No (0)
Do you have previous experience in amputation of the distal phalanx amputation for marking purposes?	Yes (1)	No (0)
Do you have ethical concerns about performing distal phalanx amputation on young animals?	Yes (1)	No (0)
Do you feel that phalanx distalis amputation is useful for marking and genotyping purposes	Yes (1)	No (0)
On which paw/feet did you find marking easiest (Front paw (FP) or hind paw (HP))?	FP	HP
Based on your experience, was the model realistic? Please comment on this regard.	Realistic	Unrealistic

## Data Availability

The data presented in this study are openly available in FigShare at DOI 10.6084/m9.figshare.25663845.

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
