# Peer review of "A 3D-Printed Dummy for Training Distal Phalanx Amputation in Mice"

_animals, 2024, doi:10.3390/ani14081253_

Round 1

Reviewer 1 Report

Comments and Suggestions for Authors

The manuscript presents the results of training of the distal phalangeal amputation (DPA) procedure in mice using realistic dummies. They were obtained using 3D printing, which is another very interesting application of these manufacturing technologies that have been attracting increasing interest in recent years. The ethical aspect of the work, i.e. increasing the efficiency of the DPA procedure without the involvement of live animal trials, should be particularly appreciated.

The results presented here describe trials with participants divided into two groups - those who train the DPA procedure on live animals (the current approach) and those who attempt to use dummies for training (the new approach proposed by the Authors). The data analysed are based on information from the questionnaires completed by the participants and the evaluation of the correctness of the execution of the DPA procedure. The statistical methods used are chosen correctly and the analysis and discussion of the results allow appropriate conclusions to be drawn.

Unfortunately, there is one, but crucial, aspect of the work that has been discussed very limitedly despite its relevance to the results obtained. This includes the description of the 3D printing procedure used, that is, the consideration and justification for the selection of a specific 3D printing method (the authors only briefly mention poly jet technology), the type of material used (the authors mention a specific material without any justification for its use) and the optimization or at least discussion of the 3D printing parameters applied (the authors only mention the 'internal procedures' of a particular company). The choice of 3D printing method and material closely relates to the quality of the produced dummies. In particular, the resolution of the 3D printing process (defined as the size of the smallest parts that can be obtained) is related to how realistically the dummy's appearance is reproduced, and the type of material is related to the weight and feel of the dummy when it is touched. The authors themselves mention how important it is to realistically reproduce the animal, but they but do not consider the possibility of choosing a 3D printing method and material. In addition, in the Introduction the possibility of using 3D printing methods in the preparation of similar dummies (or in veterinary medicine in general) is addressed by the authors in just two sentences. Therefore, there is a lack of representation of the current state of knowledge in this area.

To summarize, the manuscript cannot be considered for publication without the following corrections/additions:

1. Introduction - The current state of knowledge regarding the use of 3D printing methods in veterinary medicine, and in particular for obtaining dummies for testing various medical procedures, should be reviewed much more extensively.
2. Materials and methods - Information regarding the equipment used and the parameters of the 3D printing process should be completed.
3. Results - There should be a discussion justifying the choice of a particular 3D printing method and material and the obtained quality of the obtained mannequins. This should include consideration of the applicability of other 3D printing methods and materials - in what aspects their application may be more or less advantageous.

In addition, smaller corrections/additions should be considered:

4. The naming of mouse paws should be standardized - the terms "hind paws" (for example, Figure 2) or "rear paws" (for example, Figure 6) are used.
5. For readers less familiar with veterinary procedures, the entries in Figure 2 (90=80+10, 9=1+8, 00=3+7) may be completely incomprehensible. They should be explained in more detail.

Author Response

Author's Reply to the Review Report (Reviewer 1)

  1. Introduction - The current state of knowledge regarding the use of 3D printing methods in veterinary medicine, and in particular for obtaining dummies for testing various medical procedures, should be reviewed much more extensively.

Thank you for the comment. In the introduction, we have included two examples of the current use of dummies in teaching veterinary and human medicine (lines 82-84).

We addressed the following studies:

Akaike, M., et al., Simulation-based medical education in clinical skills laboratory. J Med Invest, 2012. 59(1-2): p. 28-35.

Cheng, A., et al., Simulation in paediatrics: An educational revolution. Paediatr Child Health, 2007. 12(6): p. 465-468.

Humpenoder, M., et al., Alternatives in Education-Rat and Mouse Simulators Evaluated from Course Trainers' and Supervisors' Perspective. Animals (Basel), 2021. 11(7).

Micallef, J., et al., Application of 3D Printing in Training Health Care Providers; the Development of Diverse Facial Overlays for Simulation-Based Medical Training. Cureus, 2022. 14(7): p. e26637.

Noyes, J.A., K.J. Carbonneau, and S.M. Matthew, Comparative Effectiveness of Training with Simulators Versus Traditional Instruction in Veterinary Education: Meta-Analysis and Systematic Review. J Vet Med Educ, 2022. 49(1): p. 25-38.

Carvalho, V.G., et al., Evaluation of 3D-Printed Dog Teeth for Pre-clinical Training of Endodontic Therapy in Veterinary Dentistry. Journal of Veterinary Dentistry, 2023: p. 08987564231210409.

McMenamin, P.G., et al., The production of anatomical teaching resources using three-dimensional (3D) printing technology. Anatomical Sciences Education, 2014. 7(6): p. 479-486.

  1. Materials and methods - Information regarding the equipment used and the parameters of the 3D printing process should be completed.

We appreciate the hint and agree that technical data is missing in the previous manuscript version. The information was updated in the materials and methods section (lines 129-140).

  1. Results - There should be a discussion justifying the choice of a particular 3D printing method and material and the obtained quality of the obtained mannequins. This should include consideration of the applicability of other 3D printing methods and materials - in what aspects their application may be more or less advantageous.

Thank you for this comment. In preliminary studies, our team measured the elasticity of five-day-old mice to select silicones to create a 3D model. To generate more haptic realism, we printed stiff molds and filled them with silicone. However, the individual toes, essential for this model, were damaged when removed from the mold. Using the Stratasys J850 Pro 3D printer, we could accurately 3D print the model using Agilus30 Clear material without losing the details of the individual toes.

Since the main objective of this study was to investigate whether using our model to learn the DPA is beneficial in laboratory animal science, we did not perform a comparative analysis of 3D printing methods. This would have resulted in higher costs for the model production and the experiment.

In addition, smaller corrections/additions should be considered:

  1. The naming of mouse paws should be standardized - the terms "hind paws" (for example, Figure 2) or "rear paws" (for example, Figure 6) are used.

Thank you for pointing this out. The term "rear paws" has been adapted and used consistently.

  1. For readers less familiar with veterinary procedures, the entries in Figure 2 (90=80+10, 9=1+8, 00=3+7) may be completely incomprehensible. They should be explained in more detail.

We appreciate this valuable comment. The number of schemes for identifying individual animals is standard in the husbandry of laboratory rodents at each animal facility, and the schemes currently in use must be displayed and learned by heart in all animal rooms. Animal caretakers and scientists must recognize their animals independently. We have optimized the numbering in the context of this study. The explanation of the numbering scheme is added in Figure 2 (lines 183-186).

Reviewer 2 Report

Comments and Suggestions for Authors

The purpose of this paper is to present a 3D-printed model for practicing distal phalanx amputation in mice, offering a lifelike and useful resource for researchers to improve their abilities in tissue sampling and marking techniques. The key findings of the research involve creating a new training approach with a 3D-printed model, which enhances the precision and effectiveness of distal phalanx amputation training. The strengths of the paper are found in its thorough methodology, such as micro-CT analysis and user feedback evaluations, showcasing the efficiency and authenticity of the 3D-printed model for training objectives.

Areas of weakness in the study include the limited sample size used, which may impact the generalizability of the findings, particularly in statistical analyses. Additionally, the study primarily focuses on the technical aspects of the amputation procedure and lacks direct examination of the mechanical properties of the dummy or potential stress experienced by the animals involved. To enhance the robustness of the research, future studies should consider larger sample sizes, investigate the impact of the training program on technical and mechanical properties, and address potential stress factors experienced by the animals during training. Furthermore, the study could benefit from including more detailed controls to compare the effectiveness of the 3D-printed dummy training method against traditional training methods or other alternatives.

In Figure 3, it would be beneficial to provide more detailed information about the participants' level of experience working with mice, such as the specific roles they hold in research (e.g., animal keepers, scientists, students) .

Line 284 mentions that users rated the dummy positively in terms of size, weight, and material. It would be helpful to elaborate on how these factors specifically contributed to the effectiveness of the training and if there were any suggestions for improvement based on user feedback 

The discussion on line 310 regarding the risks associated with lack of technical training or user nervousness could be expanded to include specific examples or case studies to illustrate these points further

The conclusion on line 297 states that the research demonstrates the potential effectiveness of using dummies as alternatives to live animals in educational and research settings. It would be valuable to discuss the long-term implications of incorporating 3D-printed dummies into standard training protocols and how this aligns with ethical considerations in animal research

How does the use of the 3D-printed dummy enhance the acquisition of procedural skills in distal phalanx amputation while aligning with the principles of the 3Rs (replace, reduce, refine) in animal research? Good be good for your paper to hight more.

Author Response

Author's Reply to the Review Report (Reviewer 2)

  1. In Figure 3, it would be beneficial to provide more detailed information about the participants' level of experience working with mice, such as the specific roles they hold in research (e.g., animal keepers, scientists, students).

We utterly agree with you that more information about the participants' professional backgrounds would be beneficial. We provide information in such regard in lines 149-152 and 227-229. Unfortunately, the low number of participants did not allow for any categorization.

  1. Line 284 mentions that users rated the dummy positively in terms of size, weight, and material. It would be helpful to elaborate on how these factors specifically contributed to the effectiveness of the training and if there were any suggestions for improvement based on user feedback

Thank you for asking about the comparability of the dummy's weight, size, and material to that of a live mouse. A 5-day-old baby mouse weighs between 3-4 g, depending on the strain. Our dummy weighs 4 g and measures 3 cm in length. Despite reaching haptic properties similar to a real mouse, which is challenging, we used rubbery material (Agilus30 Clear), which resembled realism in the cutaneous topography. We included this information, plus the participants' suggestions for improvement, in lines 292-299.

  1. The discussion on line 310 regarding the risks associated with lack of technical training or user nervousness could be expanded to include specific examples or case studies to illustrate these points further.

Thank you for pointing this out. We have removed the influence of the experimenters' nervousness and stress, as we did not measure these parameters. Regarding the lack of training, we have cited another reference; we appreciate your comment; as for future experiments, it will be helpful to include the effect of the experimenter's stress on the quality of the work on the model and animal. On the other hand, we still believe that our study clearly shows that a lack of experience leads to considerable errors in DPA performance (please see Figure 5). Please see the amendments in lines 311 – 325.

  1. The conclusion on line 297 states that the research demonstrates the potential effectiveness of using dummies as alternatives to live animals in educational and research settings. It would be valuable to discuss the long-term implications of incorporating 3D-printed dummies into standard training protocols and how this aligns with ethical considerations in animal research.

Thank you for bringing this important aspect to our attention. We have included a passage addressing this point starting from line 343-349 and 367-370.

  1. How does the use of the 3D-printed dummy enhance the acquisition of procedural skills in distal phalanx amputation while aligning with the principles of the 3Rs (replace, reduce, refine) in animal research? Good be good for your paper to hight more.

Our study aims to fulfill the 3R Reduce/Refine. The use of dummies to train the DPA reduces the number of live animals on which the training must be practiced (lines 295-302).  Figure 5 shows the error rate of participants who were asked to amputate a certain number of dead puppy mice with and without a dummy. The low stress on the animal is only maintained if the procedure is carried out correctly, as ossification and nerve innervation have not yet been completed in the last third of the phalanx. However, if more tissue is removed, it is no longer possible to refer to a minor painless procedure. Thus, dummy training and better execution on live animals also promote animal welfare (303-312). We have taken up this aspect again in lines 336 - 342.

Round 2

Reviewer 1 Report

Comments and Suggestions for Authors

Dear Authors,

Thank you for replying to the comments as well as correcting and improving the manuscript. I have no other remarks.

Best regards

Reviewer 2 Report

Comments and Suggestions for Authors

The authors revised the manuscript according to my comments and questions. I believe the manuscript can be accepted in its current state.